# Robust Texture Mapping Using RGB-D Cameras

**DOI:** 10.3390/s21093248

**Published:** 2021-05-07

**Authors:** Miguel Oliveira, Gi-Hyun Lim, Tiago Madeira, Paulo Dias, Vítor Santos

**Affiliations:** 1Institute of Electronics and Informatics Engineering of Aveiro, University of Aveiro, 3810-193 Aveiro, Portugal; mriem@ua.pt (M.O.); paulo.dias@ua.pt (P.D.); vitor@ua.pt (V.S.); 2Department of Mechanical Engineering, University of Aveiro, 3810-193 Aveiro, Portugal; 3Department of SW Convergence Technology, Wonkwang University, Iksan 54538, Korea; ghlim71@wku.ac.kr; 4Department of Electronics, Telecommunications and Informatics, University of Aveiro, 3810-193 Aveiro, Portugal

**Keywords:** texture mapping, RGB-D camera, depth consistency, face smoothing

## Abstract

The creation of a textured 3D mesh from a set of RGD-D images often results in textured meshes that yield unappealing visual artifacts. The main cause is the misalignments between the RGB-D images due to inaccurate camera pose estimations. While there are many works that focus on improving those estimates, the fact is that this is a cumbersome problem, in particular due to the accumulation of pose estimation errors. In this work, we conjecture that camera poses estimation methodologies will always display non-neglectable errors. Hence, the need for more robust texture mapping methodologies, capable of producing quality textures even in considerable camera misalignments scenarios. To this end, we argue that use of the depth data from RGB-D images can be an invaluable help to confer such robustness to the texture mapping process. Results show that the complete texture mapping procedure proposed in this paper is able to significantly improve the quality of the produced textured 3D meshes.

## 1. Introduction

RGB-D sensors have had an astounding increase in popularity on behalf of both computer graphics as well as computer vision researchers [1,2]. RGB-D sensors were initially introduced in the field of home entertainment and gaming [3]. Since then, their usage has expanded to many other areas such as robotics [4,5], agriculture [6,7], autonomous driving [8,9], human action recognition [10,11], object recognition [12,13] and 3D scene reconstruction [14,15,16,17], to name a few. Another symptom of this success is the vast number of publicly available datasets of RGB-D data, which encompass a broad range of topics, including hand action and pose dataset [18], human action dataset with thermal data [19], egocentric finger dataset [20], and grasping action dataset [21].

Texture mapping is a widely researched topic, particularly within the fields of computer graphics [22], to describe the process of transferring color collected from images, bitmaps or textures onto 3D models. More recently, the term has been used in computer vision as well, in the context of cases where color taken from photographs is mapped onto reconstructed 3D models on real scenes, e.g., [15,16,17].

A recent survey on 3D reconstruction using RGB-D cameras states that some methods achieve very high reconstruction detail [23]. Nevertheless, this work identifies some shortcomings of current approaches, in particular with respect to the appearance of the textured 3D models, stating that current approaches capture rather simplistic approximations of real-world appearance and light transport complexity. These limitations in the quality of the textures transferred to the 3D model are caused by several factors.

One important factor is the accuracy of the reconstructed 3D model. This model is usually represented in the form of a triangular mesh, although there are other alternatives, such as octrees [24], for instance. By accuracy we mean how precisely the 3D model describes the geometry of the real world scene. For example, a digital model of a wall should be a flat surface. Such models of real scenes or objects are inferred from data collected by depth sensors, which are known to be sparse, yield considerable noise, and often contain non-trivial data distortions [23]. Another common problem are scenarios that contain materials which are difficult to scan, such as glasses or mirrors [25,26]. Some authors tried to restore the depth data directly. The goal is to generate dense, noise free range data. This can be done through deep learning methods [27] or with more classical wavelet-based dictionaries [28].

Regardless, there are several approaches that produce good quality models from RGB-D data, in particular those based on signed distance fields [29,30,31], or implicit surfaces [32,33]. In this work, we use 3D meshes directly generated by a Google Tango device (https://en.wikipedia.org/wiki/Tango_(platform), accessed on 15 March 2021) [34,35,36,37], which uses a surface reconstruction approach based on dynamic spatially-hashed truncated signed distance field for mapping the scene [17]. The input 3D meshes and RGB-D images are provided by Open Constructor (https://github.com/lvonasek/tango/wiki/Open-Constructor, accessed on 15 March 2021), a software that uses a google tango device to perform 3D reconstruction.

A deeper study of the approaches proposed for geometric reconstruction from RGB-D cameras is out of the scope of the current work. Our goal is merely to establish that there are several methods which provide good quality 3D meshes but that are not entirely accurate. Since the process of texture mapping receives the 3D mesh as input, its inaccuracies may explain some of the problems encountered when texture mapping. This is especially true for hand-held cameras systems, where the drift in pose estimation is significant [15,38].

Texture mapping is the process of mapping color or texture onto a 3D model. Thus, the process requires information about the three dimensional structure of the scene, as well as the photometric properties to be mapped onto the digital 3D model.

A 3D model of a real scene or object can be produced using 3D laser scanners [39] or hand-held RGB-D cameras [17]. Consider that the 3D mesh was previously computed and is given as an input. Note that, while there are several systems which use RGB-D cameras to perform surface reconstruction, and thus build a 3D mesh of the captured scene, our approach is not limited to these systems. Yet, using an RGB-D camera for texture mapping is a forthright choice when the same RGB-D camera was used for reconstructing the 3D model [34,35,36,37]. Still, we highlight that this approach may also be of use for texturing 3D meshes produced by other systems, such as 3D LIDARs. These devices produce very accurate 3D models (e.g., see (https://velodynelidar.com/, https://www.faro.com/en-gb/products/construction-bim-cim/faro-focus/ or https://leica-geosystems.com/products/laser-scanners/scanners), accessed on 15 March 2021). However, they are expensive and cumbersome to move, which makes them ill suited to scan indoor scenarios.

One factor that may hamper the quality of the textures are inaccuracies in the camera poses estimation. Each image should have a geometric transformation describing the position of the camera when that image was collected, with respect to a common reference frame. Figure 1 shows an example of inputs containing a 3D triangular mesh and several captured images, including an RGB image, a depth image and their geometric information.

One major difference of this approach when compared with classical computer graphics applications of texture mapping is that, in this case, there are several images captured from the scene. These images should be collected with the purpose of covering the entire scene. They will often overlap with another image or even with many other images. Furthermore, this overlap may not occur in the entire region of the image which creates an intricate combination of dependencies.

This entails the existence of a mechanism that is capable of handling the duplicated photometric information, while solving the problem of information fusion. When there is more that one photometric measurement of a given surface, there are two ways of dealing with this problem: the first is to somehow fuse the information (average-based approach), and the second is to select a single measurement as representative (winner-take-all approach). Given these alternatives, the straightforward approach would be to somehow average all color contributions. However, the process of averaging the color contributions from each image results in a compromise which does not produce visually appealing results. The reason for this is that, due to the errors in the estimation of the poses of the image, the alignment between images is essentially always sub-optimal. Because of the misalignment, the average will often contain patches from different regions. Averaging these regions produces blurred textures, as is visible in Figure 2a.

The alternative approach, winner-take-all, produces visually sharp images, as shown in Figure 2b. It is also noticeable that, although the blurring is solved, the winner-take-all approach has problems as well. In particular, there are artifact which are produced due to the abrupt change in color. For example, in Figure 2b, the image that was selected to color map the triangles containing the red book was actually a poorly registered image. Because of this misalignment in the camera pose, the red color of the book is projected onto the wall behind it. This is a clear artifact which is masked (although not entirely) using an average-based approach, since the red color from the misaligned image would be averaged with wall colors provided by other images.

In summary, both approaches, average-based and winner take all, have problems. Nonetheless, we are convinced that the winner-take-all strategy has more potential to produce quality textures. This is especially true if one assumes that it is impossible to achieve a highly accurate estimation of all camera poses. To the best of our knowledge, state of the art visual SLAM approaches for hand held cameras always contain significant error (for texture mapping purposes).

The core contribution of this paper is to propose a complete processing pipeline which, taking as input a mesh and a set of inaccurately registered images, is able to produce more visually appealing textures when compared with classic winner-take-all or average-based approaches. Additionally, there are various innovative features within the pipeline, from which we highlight: (i) a propagation algorithm that ensures minimization of the transitions in the process of camera selection for each triangle in the mesh, which are responsible for creating visual artifacts; (ii) the use of RGB-D images, as opposed to simply RGB images, for the purpose of texture mapping. The additional information provided by the depth channel is used to compute what we call the depth consistency cost; (iii) a localized smoothing of texture in the regions of the mesh where there is a change in camera selection.

The remainder of the paper is organized as follows: in Section 2, related work is discussed; the problem is formulated in Section 3, and the details of the proposed approach given in Section 4. Finally, in Section 5, results are provided showing the validity of the proposed method, and Section 6 gives concluding remarks.

## 2. Related Work

To transfer color from a photograph onto a 3D mesh, the photograph must be geometrically registered w.r.t. the 3D model. Thus, another problem that affects the quality of the mapped texture is the accuracy of the estimated camera poses for mapping texture. Note that, this issue is distinct from the one described before, since the images used for texture mapping are RGB images, rather than the depth images. This is often overlooked since, in RGB-D cameras, there is a fixed rigid body transform between the RGB and the depth cameras. Nonetheless, it is important to note that misalignments in the estimated poses of the camera constitute a problem on its own [15].

Furthermore, this problem is often responsible for errors in the mapping of the texture onto the models which have a very large impact on the visual quality of the models. This is especially true when the color of a misaligned image is used to colorize a jump-edge in the 3D structure of the mesh. This effect will be described in detail in the next sections, when the proposed solution for this problem is presented.

In an attempt to address the entanglement between the quality of the geometric reconstruction and the process of texture mapping, several authors propose to conduct joint optimizations both in the geometric and photometric space. In [40], authors propose to optimize camera poses in tandem with non-rigid correction functions for all images, so that the photometric consistency of the reconstructed mapping is maximized. Reference [41] proposes a joint optimization of the geometry encoded in a signed distance field, the textures, and their camera poses along with material and scene lighting. The authors claim the joint optimization contributes to increase the level of detail in the reconstructed scene geometry and the texture consistency. In [38], the curvature of the geometric surface is considered a feature detectable by depth sensors which is incorporated as an independent quantity in every stage of the reconstruction pipeline. The authors in [42] propose to use the average shading gradient, i.e., a measure of the average gradient magnitude over all lighting directions under Lambertian shading. These measure is used as the building block for registration estimation. Others propose the usage of a reference shape-based multi-view range image registration algorithm based on rotational projection statistics features extracted from depth images. Registration is achieved by matching these features taken from the range image and the reference shape [43]. This methodology achieves state of the art results, in particular in terms of computational efficiency. The approach has also been tested in 3D object modeling problems [44]. One common problem is the lack of geometric features used to anchor the process of registration. To address this some authors proposed the usage of both external and internal object contours extracted from the RGB images. These features are aligned through a non-linear optimization procedure which is automatically initialized by a first guess generated by a linear approach [45]. Other authors avoid the texture to geometry registration problem altogether due to its complexity, casting the problem as a structure-from-motion combined with a 3D to 3D alignment problem [46].

An additional problem that must be tackled is how to make use of many available images. This problem is often referred to as multi-view texture mapping [43,45,47,48]. Two types of methodologies were proposed to tackle the problem. One approach is to blend or average the available textures in order to produce one fused texture from multiple images. The authors propose several methodologies to carry out the averaging, but all are based on some form of weighted average of the contributions of the textures in the image space [49,50]. We refer to this as an average-based approach. Average-based approaches are highly sensitive to inaccuracies in camera pose estimation. Even slight misalignments often generate ghost and blurring artifacts in the textures.

The second approach is to select, from the set of available images, a single image to be used for texture mapping. We refer to this group of methodologies as winner-take-all approaches. This methodology raises the problem of how to choose the most adequate image from a discrete set of possibilities. Moreover, the real problem is how to come up with a consistent selection of images for all the faces in the 3D mesh. Some authors propose to employ a Markov random field to select the images [51]. Others have used graph optimization mechanisms [52], or optimization procedures that minimize discontinuities between neighboring faces [53]. Winner take all approaches address the shortcomings of average-based approaches, since that the blurring and ghost artifacts are usually not present in these textures. However, in particular in regions where the selected images change, visual seam artifacts are often noticeable. The solution to this problem is to perform some form of post-processing operation. For example, Reference [54] proposes to use Poisson image editing techniques for smoothing the texture. As discussed before, both approaches have disadvantages: average-based approaches display ghosts and blurred images, and winner-take-all approaches produce sharp textures but yield artifacts containing abrupt changes in color for regions where the selected camera changes.

Finally, one other cause for incorrect texture mapping is that, due to exposure, saturation and/or low dynamic range of the CCD sensor, different images will yield different color for the same object. This is especially true for hand-held reconstructions of large buildings because, during these scanning sessions cameras are faced with very differently illuminated regions (e.g., windows, lamps, dark rooms, etc.) [17,55]. There have been several works on the topic of minimizing the photometric disparities between images. For this purpose, some authors have proposed to model the illumination and reflectance of the materials [56,57], or the vignetting effect of lenses [58,59] in the search for an absolute radiometric measure. Others approach the problem by simply attempting to alter the colors present in the images in such a way that they resemble each other, without notion of an absolute reference illumination [60,61,62].

We propose the usage of a hybrid approach: first, a prior global image adjustment stage is proposed to color correct the images in order to minimize the color differences among images. Then, a winner-take-all strategy combined with a propagation mechanism that minimizes the changes of selected cameras. Finally, a blending strategy (used only for the faces where selected images change, referred to as border faces) repairs the visual seam artifacts. Our approach contains novel pre-processing, image selection and post processing algorithms. One clear novelty of our approach is the use of depth information as a critical information to determine the selected image for texturing. Our approach directly employs depth information to conduct the process of texture mapping. It is for this reason that we describe the process as texture mapping using RGB-D cameras.

## 3. Problem Formulation

Let a triangular mesh contain *M* triangular faces, denoted as $F_{m}$. In turn, each face contains three vertex coordinates defined in $R^{3}$, as shown bellow:(1)$$\begin{array}{cc} \; & F_{m}=\left\{V_{m,i}\right\}:V_{m,i}=\left(x_{m,i},\;y_{m,i},\;z_{m,i}\right),\\ \; & \;\forall m\in \{1,M\}\wedge \forall i\in \{1,2,3\}, \end{array}$$
where *m* is the face index and *i* denotes the vertex index.

To map texture or color to a 3D mesh, a set of RGB images is required. In this work, we propose that an RGB-D camera is used for texture mapping. Thus, for each RGB image there will be a corresponding depth image. Furthermore, in order to map the color onto the mesh, information about the camera’s pose with respect to the 3D mesh reference frame is required, as well as information about the intrinsic properties of the camera. This data is denoted by a 4-tuple 〈I,D,T,K〉, which contains the RGB image I (W×H×3 matrix, where *W* and *H* are the image’s width and height, respectively), the depth image D (W×H matrix), the camera pose T (3×4 transformation matrix) and the camera’s intrinsic parameters K, (3×3 intrinsic parameters matrix). We assume that all images were taken using the same camera, and thus K is considered constant for all images. Since there is a set of *K* images, we use the superscript (k) to denote a particular RGB-D image and produce a list of 4-tuples and defined as: $\left\{\langle I^{\left(k\right)},D^{\left(k\right)},T^{\left(k\right)},K\rangle \right\}\;\;\forall \;\;k\in \left[1,K\right]$.

To colorize each of the faces in the 3D mesh, they must be projected onto the images. Section 4.1 will describe in detail the projection mechanism. For now, let us consider that $F_{m}^{\left(k\right)}$ denotes the projection of the *m*th mesh’s face ($F_{m}$) onto image *k*. The goal is to select the image which will be used to color each of the mesh’s faces. In other words, we seek the index of the selected image *s* which defines the face $F_{m}^{\left(s\right)}$ selected to map color to the 3D face $F_{m}$. The solution for the entire mesh consists of a vector of selected image indexes, one for each face $\left\{s_{1},s_{2},...,s_{M}\right\}$.

## 4. Proposed Approach

Figure 3 shows the proposed system for 3D texture mapping. The system takes in a 3D mesh, which does not contain any texture, RGB-D images, and the camera poses where the images are captured. Firstly, all 3D faces in the mesh are projected onto possible images. To discard unavailable images for faces, z-buffering and depth consistency methods are applied. From multiple cameras, which are still valid after discarding methods, only a winner camera for each face is selected. Still there is some discontinuity between faces, especially at the border faces where the indexes of cameras change, because different cameras have different properties such as white balance and exposure. In these cases, a face smoothing method is applied to reduce the discontinuity. Finally, global color correction is applied for robust texture mapping (source code is available at: https://github.com/lardemua/rgbd_tm, accessed on 27 March 2021).

### 4.1. Projection

The first step of texture mapping is the projection of the 3D faces onto the available images. We use a pinhole camera model to compute the projection of each of the face’s three vertices. The projection of a vertex can be formulated as follows:(2)$$\begin{array}{cc} \; & V_{m,i}^{\left(k\right)}=K\cdot T^{\left(k\right)}\cdot V_{m,i},\\ \; & \;\forall m\in [1,M]\wedge \forall i\in \{1,2,3\}\wedge \forall k\in [1,K], \end{array}$$
where $V_{m,i}^{\left(k\right)}=\left(x_{m,i}^{\left(k\right)},\;y_{m,i}^{\left(k\right)}\;\right)\in Z^{2}$ represents the *i*th vertex coordinates of the *m*th mesh face, projected onto the *k*th image. These coordinates are in image space, thus in $Z^{2}$. Note that the atomic structure for the projection of a face in 3D space must be a triangle, rather than the separate vertices. The reason for this is that each face may be partially visible by a camera. In order to assert a valid face projection, the face must be completely visible by the camera. Only valid faces can be used for mapping color onto the mesh. We propose to compute a projection cost $c_{p}$ which translates the validity of the projection of a given face *m* to a camera *k*. It checks if all three face vertices are projected inside the image and is expressed as:(3)$$\begin{array}{c} c_{1}\left(F_{m}^{\left(k\right)}\right)=\left\{\begin{array}{c} 1,\;\;\mathrm{if}\;\;\;\;\;\left(0\leq x_{m,i}^{\left(k\right)}\;<W\right)\;\;\wedge \;\;\left(0\leq y_{m,i}^{\left(k\right)}\;<H\right),\\ \;\forall i\in \{1,2,3\}\\ \infty ,\;\;\mathrm{otherwise} \end{array}\right.. \end{array}$$

This cost will be used later on in the process of selecting which faces to use for coloring.

### 4.2. Z-Buffering

Since each camera views a 3D model from any arbitrary pose, it is possible that the model contains faces which are not visible by one particular camera. There are two possible reasons for this: the face may not be within the camera view fustrum, this was covered in Section 4.1, or it may be occluded by another face. This occurs when the occluding face intersects the line of sight of the camera-occluded face. The occluding face is closer to the camera than the occluded face. We refer to this as a face-based z-buffering, since this problem is very similar to the z-buffering [63]. The difference here is that, once again, the atomic structure are the faces. In other words, the occlusions must be computed for triangles rather than pixels, as is the case of classical z-buffering. An occluded face will not contain correct color information and thus must be discarded from the face selection process used for texture mapping. To do this we propose to compute the z-buffering cost $c_{z}$ as follows:(4)$$\begin{array}{c} c_{2}\left(F_{m}^{\left(k\right)}\right)=\left\{\begin{array}{c} \infty ,\;\;\mathrm{if}\;\;\;\overset{intersection}{\overset{︷}{\left(S_{1}\left(F_{m}^{\left(k\right)}\right)\vee S_{2}\left(F_{m}^{\left(k\right)}\right)\right)}}\;\wedge \overset{\mathrm{dist}\;\mathrm{to}\;\mathrm{camera}}{\overset{︷}{S_{3}\left(F_{m}^{\left(k\right)}\right)}}\\ \;\forall m\in [1,M],\\ 1,\;\;\mathrm{otherwise} \end{array}\right., \end{array}$$
where $S_{1}$, $S_{2}$ and $S_{3}$ are logical statements, the first two concerning the intersection of one face by another, while the latter pertains to the distance between the faces and the camera. Equation (Equation 4) asserts whether $F_{m}^{\left(k\right)}$ is occluded. For that, all other faces in the mesh are tested. We use alternative subscripts n,j (instead of m,i) to denote the (possibly) occluding faces and thus use the notation $F_{n}^{\left(k\right)}\;\;\forall n\in \left[1,M\right]\wedge n\neq m$. Statement $S_{1}$ verifies if there is a vertex from a face $F_{n}^{\left(k\right)}$ which lies inside face $F_{m}^{\left(k\right)}$:(5)$$\begin{array}{c} S_{1}\left(F_{m}\right)≜[\forall n\in \left[1,M\right],n\neq m,\forall j\in \left\{1,2,3\right\}\;\;\;\exists \;\;V_{n,j}:\\ \;\;\;\min\left(\left\{u_{\{m,n\},j},v_{\{m,n\},j},w_{\{m,n\},j}\right\}\right)<0], \end{array}$$
where [·] denotes the Iverson bracket, *u*, *v* and *w* are the face $F_{m}$ based barycentric coordinates of vertex $V_{n,j}$ defined as:(6)$$\begin{array}{c} u_{\{m,n\},j}=\frac{\left(y_{m,2}-y_{m,3}\right)\left(x_{n,j}-x_{m,3}\right)+\left(x_{m,3}-x_{m,2}\right)\left(y_{n,j}-y_{m,3}\right)}{\left(y_{m,2}-y_{m,3}\right)\left(x_{m,1}-x_{m,3}\right)+\left(x_{m,3}-x_{m,2}\right)\left(y_{m,1}-y_{m,3}\right)} \end{array}$$(7)$$\begin{array}{c} v_{\{m,n\},j}=\frac{\left(y_{m,3}-y_{m,1}\right)\left(x_{n,j}-x_{m,3}\right)+\left(x_{m,1}-x_{m,3}\right)\left(y_{n,j}-y_{m,3}\right)}{\left(y_{m,2}-y_{m,3}\right)\left(x_{m,1}-x_{m,3}\right)+\left(x_{m,3}-x_{m,2}\right)\left(y_{m,1}-y_{m,3}\right)} \end{array}$$(8)$$\begin{array}{c} w_{\{m,n\},j}=1-u_{\{m,n\},j}-v_{\{m,n\},j} \end{array}$$

Statement $S_{2}$ searches for an intersection between the edges of (possibly) occluded face $F_{m}$, defined by the vertices $V_{m,i}$ and $V_{m,i+1}\;\;\;\forall i\in \left\{1,2,3\right\}$, with any of the edges of (possibly) occluding face $F_{n}$, defined by the vertices $V_{n,j}$ and $V_{n,j+1}\;\;\;\forall j\in \left\{1,2,3\right\}$. It is defined as follows:(9)$$\begin{array}{cc} S_{2}\left(F_{m}\right)≜ & [\forall n\in \left[1,M\right],n\neq m,\forall i,j\in \left\{1,2,3\right\},\exists \;\;a_{\left\{m,n\},\{i,j\right\}}:\\ \; & \min\left(x_{m,i},x_{n,j}\right)\leq a_{\left\{m,n\},\{i,j\right\}}\leq \max\left(x_{m,i+1},x_{n,j+1}\right)\\ & \wedge \;\left(y_{m,i}-y_{m,i+1}\right)\cdot \left(x_{n,j}-x_{n,j+1}\right)\neq \\ \; & \;\;\;\left(y_{n,j}-y_{n,j+1}\right)\cdot \left(x_{m,i}-x_{m,i+1}\right)], \end{array}$$
where *a* is computed as follows:(10)$$\begin{array}{c} a_{\left\{m,n\},\{i,j\right\}}=\frac{x_{m,i}\left(x_{n,j}\left(y_{m,i+1}-y_{n,j+1}\right)+x_{n,j+1}\left(y_{n,j}-y_{m,i+1}\right)\right)}{b_{\left\{m,n\},\{i,j\right\}}}+\\ \;\frac{x_{m,i+1}\left(x_{n,j}\left(y_{n,j+1}-y_{m,i}\right)+x_{n,j+1}\left(y_{m,i}-y_{n,j}\right)\right)}{b_{\left\{m,n\},\{i,j\right\}}}, \end{array}$$
and *b* is determined by Equation (Equation 11):(11)$$\begin{array}{c} b_{\left\{m,n\},\{i,j\right\}}=\left(x_{m,i}-x_{m,i+1}\right)\left(y_{n,j}-y_{n,j+1}\right)+\\ \;\left(x_{n,j+1}-x_{n,j}\right)\left(y_{m,i}-y_{m,i+1}\right). \end{array}$$

Finally, statement $S_{3}$ verifies if the occluding face $F_{n}$ is closer to the camera than the occluded face $F_{m}$. An occlusion is acknowledged when a vertex from the occluding face is closer to the camera than the vertex from the occluded face which is closest to the camera:(12)$$\begin{array}{cc} S_{3}\left(F_{m}\right)≜ & [\forall n\in \left[1,M\right],n\neq m,\forall i,j\in \left\{1,2,3\right\}\;\;\;\exists \;\;V_{n,j}:\\ \; & \;D\left(x_{n,j},y_{n,j}\right)<\min_{i}\left(D\left(x_{m,i},y_{m,i}\right)\right)], \end{array}$$
where D(x,y) denotes the range measurement collected from the depth image for pixel x,y (see Section 3).

Figure 4 shows snap shots during z-buffering. Many faces, which must come from the gray box or the table, are hidden because of the faces from the red book. The book is the closest object from the perspective of the camera, in views where the book is visible (see the video: https://youtu.be/X9jk-K2q2cY, accessed on 15 March 2021).

The result of z-buffering for a camera is shown in Figure 5. The blue triangles represent all visible faces from the camera view, while the black ones represent hidden faces, which may be projected from other cameras.

### 4.3. Depth Consistency

The main contribution of this paper is to propose the usage of RGB-D images, as opposed to simply RGB images, for the purpose of texture mapping. The additional information provided by the depth channel of RGB-D images is used to compute what we call the depth consistency cost.

The idea is to compare two different values of depth for each projected face $F_{m}^{\left(k\right)}$, in order to assess if that face contains incorrect photometric information. Note that we do not mean incorrect in the sense that the photograph was somehow corrupted, but rather in the sense that the photometric information is inconsistent with the 3D model. Incorrect photometric information occurs due to a misalignment between the camera and the 3D model. This is caused by inaccurate camera pose estimations, which is a problem that is present in most 3D reconstructions. Furthermore, this inconsistency between the image and the 3D mesh is noticeable mostly in corners and jumps in the 3D structure. In some cases a slight misalignment is innocuous for texture mapping, while in others it may be critical. Our approach consists in comparing the depth computed by the projection of a 3D face onto the image with the depth measured by the depth sensor. Hence the need for an RGB-D camera. A large difference between these indicates that the face viewed by the depth camera is not the same face computed by the projection (due to the misalignments). In such cases, since the depth and RGB images are registered, we may also conclude that the color information contained in the RGB image is not from the 3D face and thus cannot be used for texture mapping.

We start by computing the depth for each vertex *i* of a given face *m*. We refer to this depth *e* as the estimated depth because it is computed after the $L_{2}$ norm of the vertex coordinates transformed into the camera coordinate frame, thus:(13)$$e_{m,i}^{\left(k\right)}=\left\|T^{\left(k\right)}\cdot V_{m,i}\right\|,$$
where ‖·‖ denotes the $L_{2}$ norm. The second measurement used is the depth measured directly from the depth sensor. We refer to this depth *d* as the measured depth. It is taken directly from the depth image at the coordinates computed for the projected vertex computed from Equation (Equation 2) (see Section 4.1):(14)$$d_{m,i}^{\left(k\right)}=D^{\left(k\right)}\left(x_{m,i}^{\left(k\right)},y_{m,i}^{\left(k\right)}\right),$$
and, finally, the depth consistency is asserted by computing the cost $c_{d}$ as follows:(15)$$c_{3}\left(F_{m}^{\left(k\right)}\right)=\left\{\begin{array}{c} 1,\;\mathrm{if}\;\max_{i}\left(\left|e_{m,i}^{\left(k\right)}-d_{m,i}^{\left(k\right)}\right|\right)<t_{d}\\ \infty ,\;\;\;\;\;\;\;\mathrm{otherwise} \end{array}\right.$$
where $t_{d}$ is the depth consistency threshold, which establishes the maximum allowed inconsistency between these two measurements. Using Equation (Equation 15) it is possible to discard textures from images which are not consistent with the 3D model that is going to be textured.

Figure 6 shows the sensitivity of depth consistency with different thresholds ($t_{d}$): 2, 5, 10 and 15 cm. The gray triangles represent inconsistent faces with a depth consistency threshold, while blue and black triangles represent visible and hidden faces, respectively. Since it is reported that the distance error of the sensor is around 3 cm, there are many depth-inconsistent faces in Figure 6a, which uses 2 cm as the threshold.

Figure 7 shows a textured mesh produced using the depth consistency mechanism. Without this mechanism, a misaligned camera projects red color into the wall (see Figure 2b). With our approach, since the misalignment causes a depth inconsistency, the misaligned image is not selected for texture mapping which avoids the artifacts.

### 4.4. Camera Selection

As discussed in Section 3, the goal of texture mapping using multiple cameras and a winner-take-all approach is to select, for each face in the 3D model, the image which is used to colorize that face. The process of colorization (or texture mapping) of a triangular face $F_{m}$ using a single selected image index $s_{m}$ is relatively straightforward: the projection of the 3D face, denoted as $F_{m}^{\left(s_{m}\right)}$, is cropped from image $s_{m}$ and copied onto the textured model. We do this by annotating the uv coordinates and the selected image path into the *.obj file which will contain the 3D textured model. The uv coordinates are computed after the projection of the 3D vertices onto the selected image coordinates as detailed in (Equation 2), normalized by the image size:(16)$${\left[\begin{array}{c} u\\ v \end{array}\right]}_{m,i}^{\left(s_{m}\right)}={\left[\begin{array}{c} \frac{x}{W}\;\\ \frac{y}{H} \end{array}\right]}_{m,i}^{\left(s_{m}\right)}\;\forall m\in \left[1,M\right]\wedge \forall i\in \left\{1,2,3\right\}.$$

In other words, the core concern is that of selecting the image which is used to map color. After this, the process is straightforward. The image selection process is based on the computation of a per image global cost $c\left(F_{m}^{\left(k\right)}\right)$ associated with each face.
(17)$$c\left(F_{m}^{\left(k\right)}\right)=\prod_{l=1}^{4}c_{l}\left(F_{m}^{\left(k\right)}\right),$$
where $c_{1}$, $c_{2}$ and $c_{3}$ correspond to the costs associated to the projection (Section 4.1), the z-buffering (Section 4.2), and the depth consistency (Section 4.3), respectively. Since these three costs return one of two values 1 or *∞* (see equations. Equations (Equation 3), (Equation 4) and (Equation 15)), they are used to assess whether the image can be used or not to colorize a given face. If any of these criteria fail, the cost takes value *∞*, which means the image cannot be used to colorize the face. If all three criteria return value 1, then we refer to this projected face as an available face, meaning that it is a projected face that can be used to colorize the 3D face. Note, however, that the three costs discussed above do not make distinctions between available faces.

Cost $c_{4}$, in turn, is meant to provide a measure of the quality of an image, for a given 3D face. This quality will be the deciding factor in the selection of one image, i.e., one projected face, from the set of available faces. Bear in mind that this process of selection has a very large impact on the quality of the textures observed in the 3D model. We propose to test four possibilities for the cost $c_{4}$. We refer to these as the different approaches for camera selection. The following sections will refer four approaches: random selection, which is used only for benchmarking the other approaches, last camera index, in which the cameras with the highest index (the last ones) are preferred (have lower $c_{4}$), largest projected face area, which privileges the cameras which contain the largest triangle projection (this may be interesting to maximize the resolution of the produced textures), and finally, our approach, which runs an iterative propagation, defining cost $c_{4}$ contextually, so that when a parent face propagates into a child face, the lowest cost $c_{4}$ for the child face will be achieved if the child face selected camera is the same as the parent face selected camera, as shown in the video: https://youtu.be/7j-bukidACs, accessed on 15 March 2021). The propagation starts from a random face in the mesh and is carried out by visiting the neighboring faces of the previously visited faces. The algorithm annotates which faces have already been visited in order to cycle through the entire mesh, without revisiting any. Each time a face is visited, the camera selected to texture map it is the same as that of the neighbor triangle from which the visit was triggered, i.e., the parent triangle. Note that sometimes this is not possible because the visited triangle is not observed by the camera selected for the parent triangle. When this occurs, transitions in camera selection will be noted in the mesh.

### 4.5. Border Face Smoothing

As discussed in Section 2 the usage of a winner-take-all approach results in visual artifacts in the texture seams, i.e., the regions of the mesh where the selected faces change display abrupt color transitions. We annotate a face as a border face if it contains a neighbor face with a different selected camera. Let δ(·) be a function that determines if a face is a border face or not:(18)$$\delta \left(F_{m}^{\left(s_{m}\right)}\right)=\left\{\begin{array}{c} 1,\;\mathrm{if}\;s_{m}\neq s_{\beta_{o}\left(m\right)}\\ 0,\;\;\;\;\;\;\;\mathrm{otherwise} \end{array}\right.,\;\forall o\in \left\{1,2,3\right\},$$
where $\beta_{o}\left(m\right)$ is a function that retrieves the *oth* neighbor of face *m*.

Figure 8 (top) show a mesh where each color represents a selected camera index. Two criteria for image selection are shown: random (a) and (c); propagation (b) and (d). The faces identified as border faces are painted yellow in Figure 8 (bottom). As expected, the random selection criteria contains many transitions between selected cameras (see Figure 8a). As a result, there is a great number of border faces in Figure 8c). Inversely, the propagation approach contains far less transitions (Figure 8b), which leads to a reduced number of border faces (Figure 8d).

The idea of the proposed smoothing mechanism is to conduct an average-based approach only on the border faces. We expect that this blending will mitigate the visual seam artifacts typical of winner-take-all approaches. Figure 9 shows an example of the proposed smoothing mechanism. Let a given face *F* be a boundary face in 3D and the three neighbor faces of *F* have a selected camera index, $n_{0}$, $n_{1}$ and $n_{2}$. The projection of *F* onto each of the neighbor face selected cameras results in projected faces $F^{n_{0}}$, $F^{n_{1}}$ and $F^{n_{2}}$, as shown in Figure 9e–g, respectively. Since these are projections of the same 3D face to different cameras, the shape of these faces may vary, as shown in Figure 9h–j. To make averaging the color of these faces possible, we propose to warp the triangular faces $F^{n_{1}}$ and $F^{n_{2}}$ to match the shape of $F^{n_{0}}$. Let $\omega^{\;\;i}\left(\cdot \right)$ be a warping function that creates a new face with the same shape as *i* by means of linear interpolation warping. Then, one may express the value of pixels (u,v coordinates) in the new averaged face A as a weighted average:(19)$$\begin{array}{c} A\left(u,v\right)=w_{n_{0}}\left(u,v\right)\cdot F^{n_{0}}\left(u,v\right)+w_{n_{0}}\left(u,v\right)\cdot \omega^{\;\;n_{0}}\left(F^{n_{1}}\left(u,v\right)\right)\\ +w_{n_{2}}\left(u,v\right)\cdot \omega^{\;\;n_{0}}\left(F^{n_{2}}\left(u,v\right)\right) \end{array}$$
where w(u,v) is a weight mask computed based on the sum of the barycentric coordinates of the two vertices that are contained by each neighboring face edge. We use barycentric coordinates to compute the weight masks, as shown in Figure 9k–m since they are known to yield the best results for interpolating texture inside triangles. The formulation for barycentric coordinates was presented in Section 4.2, Equation (Equation 6), and will therefore not be introduced once more. Figure 9c,d show the current given face (*F*) and the averaged result, respectively. Since the $n_{0}$ textured face ($F^{n_{0}}$) whose neighbor face comes from the right upper is brighter than the other two textured faces ($F^{n_{1}}$ and $F^{n_{2}}$), the right upper side of the averaged result is slightly brighter than the other sides.

### 4.6. Global Color Adjustment

Section 4.5 described the mechanism proposed for blending the colors of several camera in the border faces. The approach significantly enhances the visual appearance of the textured mesh by averaging the color of the different cameras in a weighted fashion. Nonetheless, the border face smoothing approach is always limited to finding a common ground between different colors proposed by the cameras. In other words, the smoothing mechanism is designed to compensate for minimal color differences. As such, if these colors are considerably distinct, the smoothing mechanisms will not be sufficient.

Global color adjustment is an optional part of the pipeline. When used, this global color correction procedure adjusts the colors of all the images before the image selection, to texture map the triangles in the mesh. As such, it occurs as a pre-processing operation, before the texture mapping process.

The goal is to adjust the colors of the images which will later be used for texture mapping, in a way that color inconsistencies are minimized. The set of parameters θ to optimize are the bias and scaling in the three color channels of all available images. We use the YCbCr colorspace since this color channel is known to yield less cross channel artifacts when compared to the standard RGB colorspace [61,62]. We define the cost function *f* as the sum of color differences between projections of the mesh’s vertices to different cameras:(20)$$\begin{array}{c} f≜\sum_{\{k1,k2\},o}g\left(I^{\left(k1\right)}\left(V_{o}^{\left(k1\right)},\theta \right),I^{\left(k2\right)}\left(V_{o}^{\left(k2\right)},\theta \right)\right),\\ \forall k1,k2\in [1,K],k2>k1,o\in [1,N] \end{array}$$
where *K* is the number of cameras, *N* the number of mesh vertices, $I^{\left(k1\right)}\left(V_{o}^{\left(k1\right)},\theta \right)$ is the k1 color adjusted image using parameters θ at pixel coordinates $V_{o}^{\left(k1\right)}$, and g(·) is a function that returns the color difference between two pixels.

Using Equation (Equation 20) we are able to adjust the colors of a set of images so that, when the optimization finalizes, the color consistency throughout the entire dataset is maximized. In this video (https://youtu.be/t2jNJwag_ng, accessed on 15 March 2021) it is possible to observe the optimization routine as it proceeds to color correct the image set.

## 5. Results

This section presents results showing that our proposed approach is capable of significantly reducing the visual artifacts in texture meshes. Since partial results were presented in Section 4, during the detailed presentation of the proposed algorithms, the focus of this section is to evaluate the overall performance of the proposed system.

The first focus is on camera selection approaches. As discussed in Section 4.4, we have tested four different camera selection criteria. Figure 10, Figure 11, Figure 12 and Figure 13 show the results of using camera selection criteria: random, last camera index, largest projected face area and propagation, respectively. For visualization purposes, in this experiment, we have used a subset of only six camera images, k={0,7,14,21,28,35}. Costs for camera selection criteria are computed for each face using projection, z-buffering, and depth consistency costs, in addition to cost $c_{4}$, corresponding to the approach in question (see Equation (Equation 17)). The color coding for those figures is the following: occluded faces, i.e., where $c_{2}\left(F_{m}^{\left(k\right)}\right)=\infty$ have black color; faces where depth is inconsistent, i.e., where $c_{3}\left(F_{m}^{\left(k\right)}\right)=\infty$ have grey color; available faces, i.e., where $c_{1}\left(F_{m}^{\left(k\right)}\right)\cdot c_{2}\left(F_{m}^{\left(k\right)}\right)\cdot c_{3}\left(F_{m}^{\left(k\right)}\right)=1$, are colored white; additionally, selected faces, where $F_{m}^{\left(s_{m}\right)}:s_{m}=k$, are colored green.

Figure 10 shows the results of selected faces with random criteria. As expected in this case, the selected faces (green) are randomly scattered throughout all available images.

Figure 11 shows the results of selected faces using the largest camera index criteria. In this case, the camera with the largest index will have the smallest cost. As such, all available faces in Figure 11 (*f*) (k=35) were selected. Additional faces, which are not seen or are occluded from k=35, are selected from other cameras. For example, because camera k=14 is further away from the mesh, it can view faces on the wall that are not visible by k=35, and these faces have selected camera k=14.

Figure 12 shows the results obtained for largest projected triangle area. In this case, it is also possible to see that the larger triangles are the ones selected. For example, the camera which is closest to the red book, k=21 will have larger projected triangles (since the intrinsics for all cameras are the same), and therefore contains many selected faces.

Figure 13 shows the results of face selection using the proposed propagation criteria. The random initialization selected camera k=0 and a mesh face near the red book as the propagation seed. The mechanism will then iteratively propagate from one face to another, avoiding a change in the selected camera whenever possible. As expected, this resulted in all available faces from k=0 being selected. Then camera k=7 is used to colorize the top left faces in the wall (which are not viewed by k=0), and k=14 is selected to colorize the faces from the table which are also not visible from k=0. We can see that the algorithm creates large patches of connected faces with the same selected camera. This was the intended effect, since it should lead to an enhancement in the quality of the textured meshes.

Figure 14 shows the final result, that is, the textured meshes obtained using each of the camera selection approaches. In addition to those, results for the border face smoothing and global image adjustment are also depicted. Figure 15 displays a zoomed-in region (the blue rectangle in Figure 14a) of the images shown in Figure 14. Considering first only the 4 alternatives for camera selection criteria (Figure 14a,d) we can say that, as a general guideline, the changes in selected camera seem to always create visual artifacts. The extreme case is the random approach, which contains many transitions between selected cameras, resulting in a very unappealing textured mesh Figure 14a. The proposed approach produces the best quality textures of all four approaches (see Figure 14d. Despite this, some visual seam artifacts are still visible. The border face smoothing algorithm repairs the abrupt visual transitions, which results in even higher quality textures (see Figure 14e. In addition, the complete proposed system contains the global color adjustment approach. As seen in Figure 14f, it produces the best textured meshes of all. Texture seam artifacts are much less noticeable than in any of the other approaches. This asserts the validity of the proposed approach to perform texture mapping. Our approach is particularly robust to capture sessions where the cameras are only roughly aligned.

## 6. Conclusions

This work proposed a novel approach for texture mapping of 3D models. The quality of the textured mesh may be hampered by inaccuracies in the geometric representation of the scene (the 3D model), in the estimation of the pose of the cameras that produced the images to be used in the texture mapping process. Finally, because there are several overlapping images of the scene, color inconsistencies in those images caused by illumination, auto exposure etc., also contribute to produce visually unappealing artifacts in the textured meshes.

We propose to address the problem by first performing a set of verifications for each pair of camera and projected face, in order to assess if the camera is usable by the face for texture mapping. Our approach takes into account both the triangle-based z-buffering, as well as the depth consistency to carry out this verification. Then, a propagation mechanism, similar to a region growing algorithm is used to minimize the transitions between selected cameras, since these are the regions more prone to display those visual artifacts. Besides minimizing the number of border faces, our approach also conducts a blending procedure which is exclusively used for them. Finally, a global color homogenization procedure is introduced, that adjusts the image colors in order to increase the color similarity between images that view the same regions of the scene.

Results show that our system is capable of producing visually appealing textured meshes. In particular, we show that each of the individual mechanisms contributes to enhance the produced textures. We compare our propagation approach for camera selection with other three approaches. Finally, results also have shown how the border face smoothing and the global color adjustment enhance the visual appearance of the textured mesh.

For future work, we plan to incorporate all these proposed mechanisms into a single, joint optimization procedure capable of producing even higher quality textures. In addition, we will look into how deep learning approaches may be useful to address, in a non heuristic manner, some of the problems discussed in this paper.

## Figures and Tables

**Figure 1 sensors-21-03248-f001:**
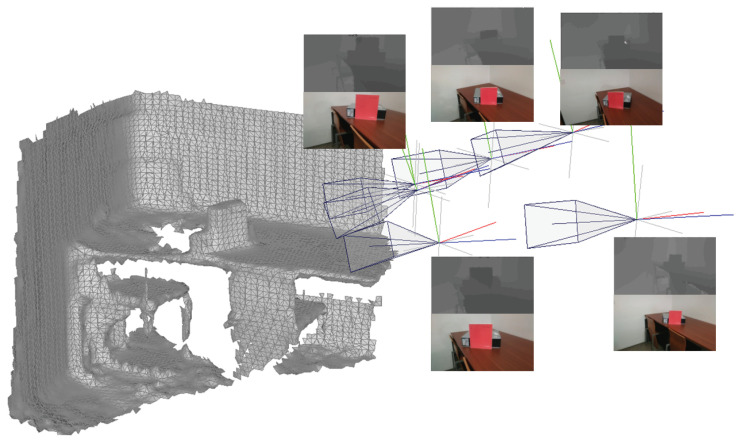
Inputs for 3D texture mapping.

**Figure 2 sensors-21-03248-f002:**
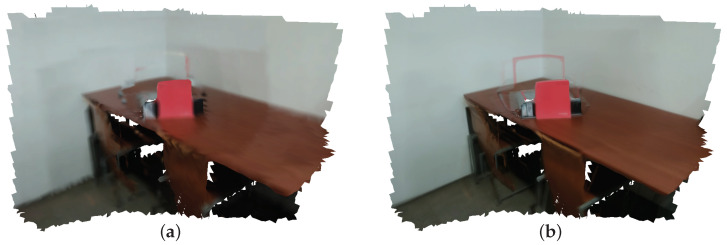
Using one or many images in texture mapping: (**a**) average-based approach; (**b**) winner-take-all approach. Images produced by OpenConstructor before final optimization (**a**) and after optimization (**b**).

**Figure 3 sensors-21-03248-f003:**
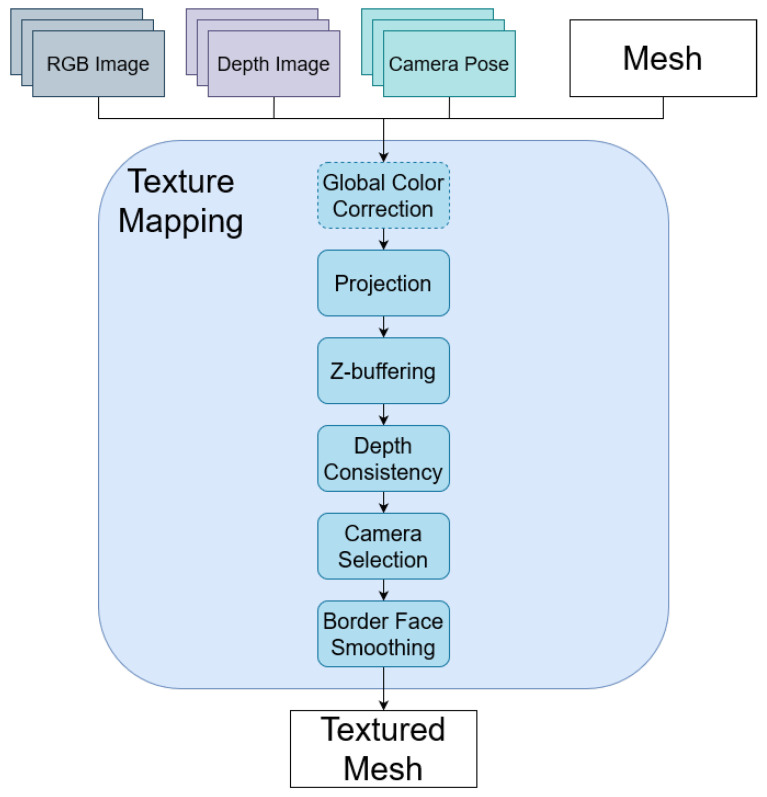
Proposed system diagram.

**Figure 4 sensors-21-03248-f004:**
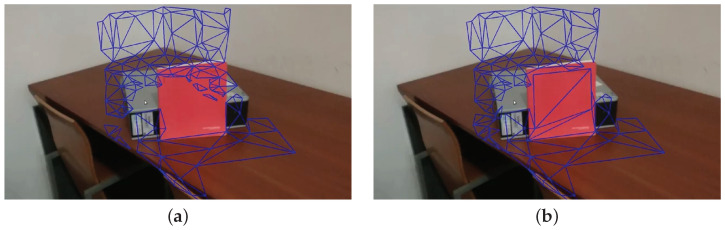
Instantaneous snapshots of the z-buffering procedure: (**a**) before processing the faces on the red book, the faces behind the book are included in the list of non-occluded faces; (**b**) the faces containing the red book are processed and, since they occlude the faces discussed in (**a**), these are removed.

**Figure 5 sensors-21-03248-f005:**
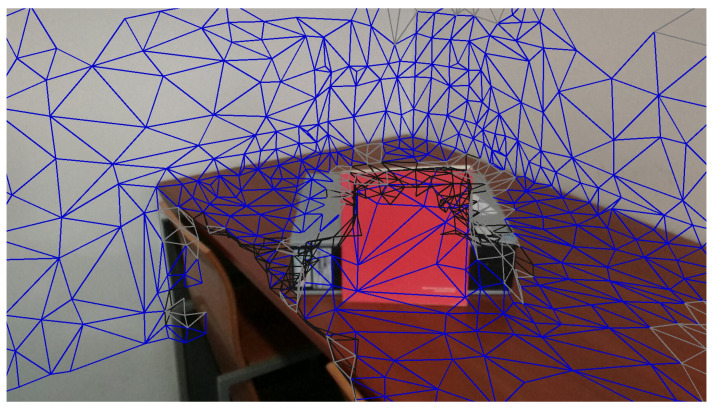
Face-based z-buffering. Black triangles are triangles detected as occluded.

**Figure 6 sensors-21-03248-f006:**
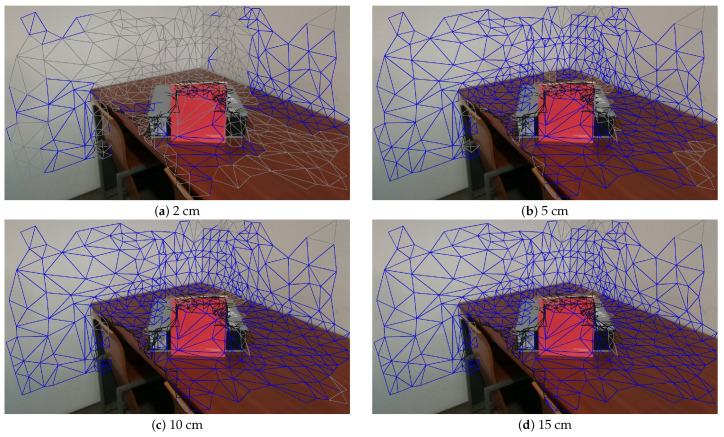
Sensitivity of depth consistency with different thresholds.

**Figure 7 sensors-21-03248-f007:**
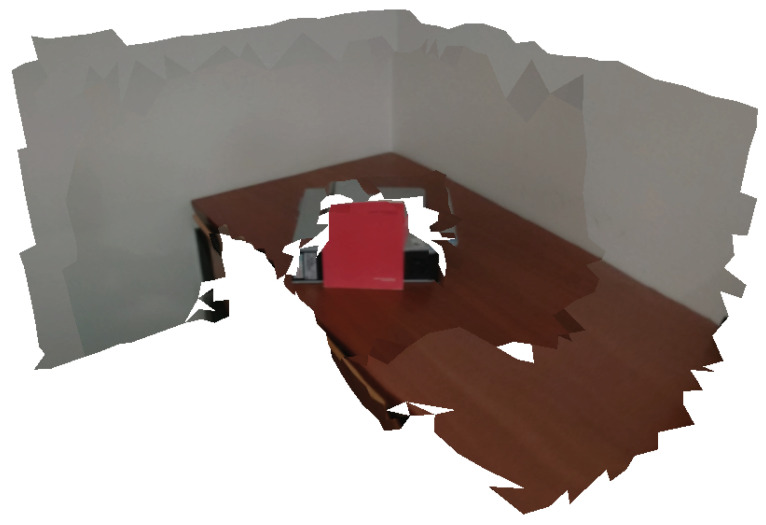
Textured mesh generated using the depth consistency algorithm. Notice that, in comparison with the standard winner-takes-all strategy displayed in Figure 2b, the red color (from the book) is not mapped on the wall.

**Figure 8 sensors-21-03248-f008:**
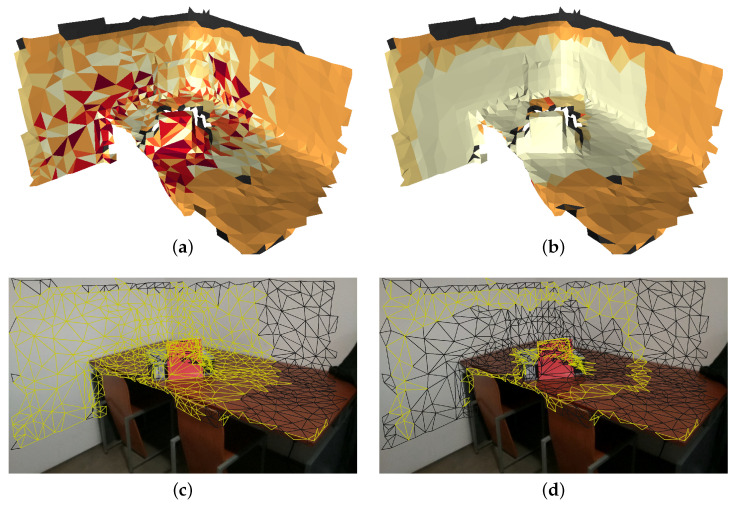
The meshes colorized according to the selected camera are shown for the random criteria (**a**) and the propagation criteria (**b**). Results of border face detection (yellow painted faces) using random criteria for camera selection (**c**); propagation criteria for camera selection (**d**).

**Figure 9 sensors-21-03248-f009:**
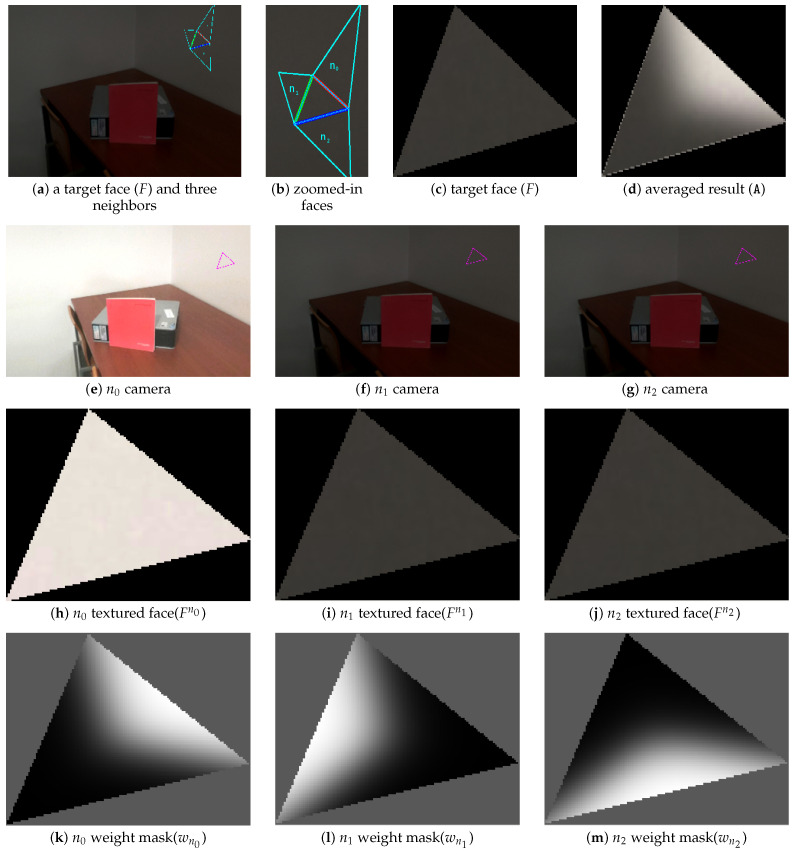
Example of the proposed smoothing mechanism.

**Figure 10 sensors-21-03248-f010:**
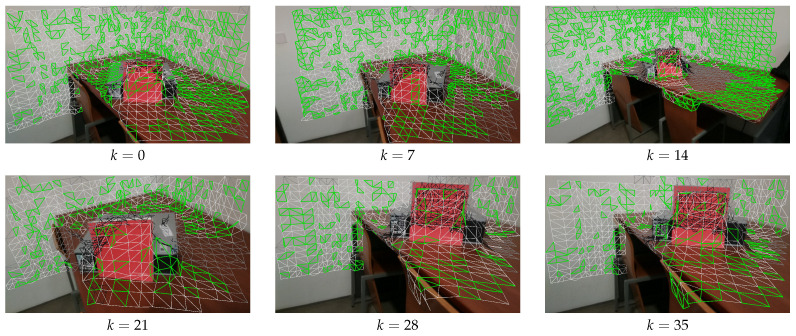
Random criteria for selecting the faces; Color notation: black, occluded faces; gray, depth inconsistent faces; white, available faces; green, selected faces.

**Figure 11 sensors-21-03248-f011:**
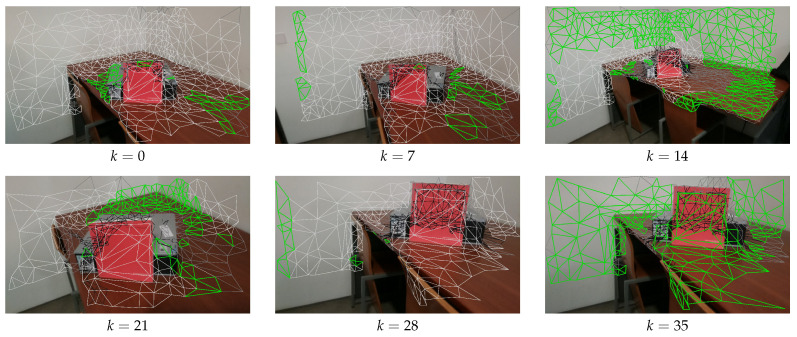
Largest camera index criteria for selecting the faces; Color notation is the same as in Figure 10.

**Figure 12 sensors-21-03248-f012:**
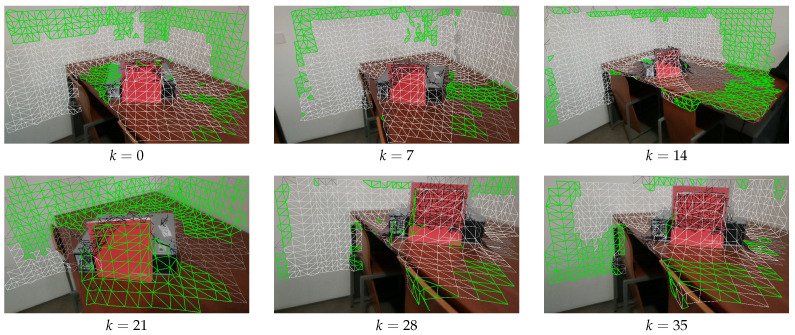
Largest projected face area criteria for selecting the faces; Color notation is the same as in Figure 10.

**Figure 13 sensors-21-03248-f013:**
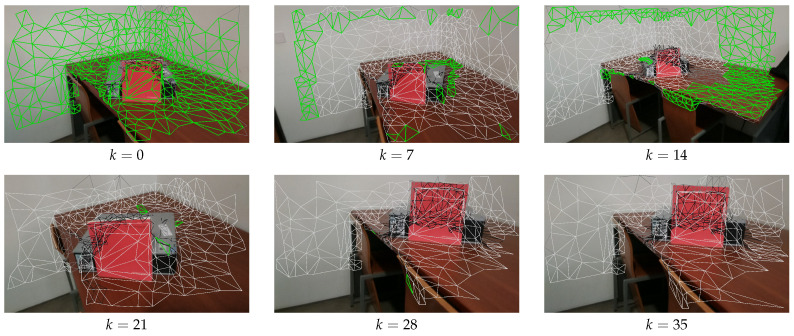
Propagation criteria for selecting the faces; Color notation is the same as in Figure 10.

**Figure 14 sensors-21-03248-f014:**
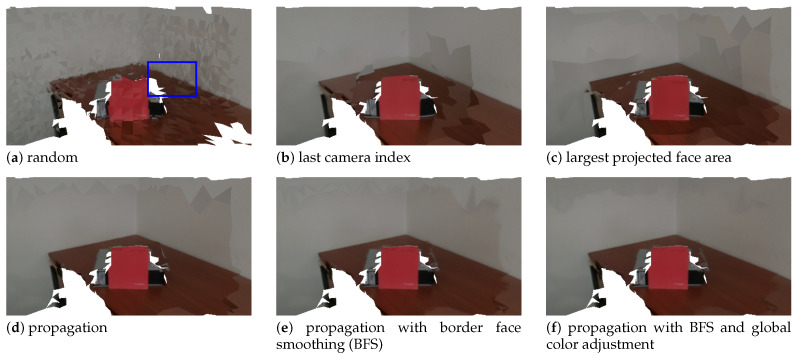
Textured meshes obtained using distinct camera selection approaches, as well as border face smoothing and global color adjustment additional processing.

**Figure 15 sensors-21-03248-f015:**
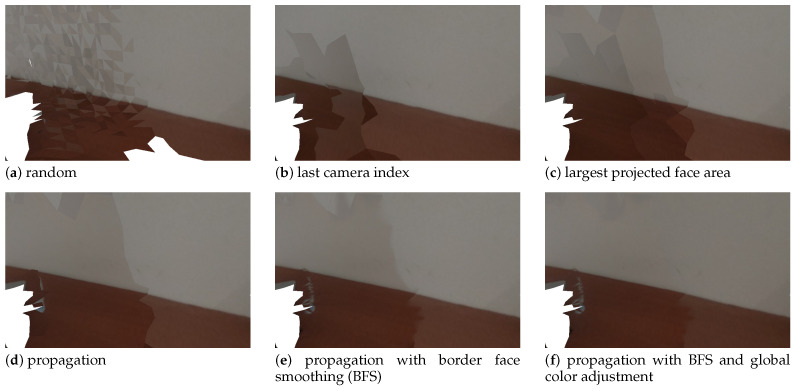
Zoom in of the area shown by the blue rectangle of Figure 14 for each respective approach.

## Data Availability

Not applicable.

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
