# Peer review of "Robust Texture Mapping Using RGB-D Cameras"

_sensors, 2021, doi:10.3390/s21093248_

Round 1

Reviewer 1 Report

The authors present a novel approach to texture mapping using RGB-D data, i.e., RGB images and corresponding depth images. Overall, the work is clearly presented. Even though it is a a work in progress, the results are promising. It would be interesting to see the results shown on another scene as well, especially a cluttered scene.
Line 130:  "In an attempt address" should read "In an attempt to address"

Author Response

“The authors present a novel approach to texture mapping using RGB-D data, i.e., RGB images and corresponding depth images. Overall, the work is clearly presented. Even though it is a work in progress, the results are promising."

We thank the reviewer for the positive feedback.

“It would be interesting to see the results shown on another scene as well, especially a cluttered scene."

We agree with the reviewer that it would be interesting to provide more results. However, in this phase of our work, our goal was to present a novel approach and validate the proposed processing pipeline with a proof of concept. The acquisition and processing of scenes has been negatively impacted by the current pandemic. It is definitely part of our future work, however, we believe that the examples presented in this paper, although limited, are mature enough to demonstrate the relevance, contributions and main features of our work. We are currently working in the area of mesh quality evaluation, defining and developing several qualitative and quantitative metrics that we expect to, in the future, use to compare various mesh processing techniques with the one presented in this manuscript, leveraging different scenes.

"Line 130:  "In an attempt address" should read "In an attempt to address“

We have corrected this error. We thank the reviewer for pointing it out.

Reviewer 2 Report

Authors propose a method for mapping texture into a 3D mesh, by utilizing RGB-D  data in order to compensate for errors in camera-pose estimations that characterize regular RGB images that lead to unwanted visual artifacts. The proposed method consists of typical stages where the 3D mesh is projected onto the camera’s plane for each acquisition, and a z buffering stage is used to define which components of the mesh are visible. Consequently, a stage where the depth information is used to validate each mesh component, by comparing the estimated and measured depth, and a selection of the best camera for each triangle is performed via a propagation algorithm that promotes the selection of same cameras for neighboring triangles. The border faces (triangles that are neighboring to others with different cameras selection) are blended in order to smooth the transitions, and the images are color-corrected in a global optimization process that minimizes the inconsistency between the color of same faces as appear to different cameras.

The paper is fairly structured and the language is good. The algorithmic flow seems straightforward, but I have doubts regarding the ease of reproducibility if authors do not provide source code in a public repository, since the overall algorithm contains numerous technicalities.

Authors should make clear what they propose and what is going to be their contribution in the introduction of the paper, because in it’s current form is rather vague. The contribution becomes clear only on line 260, pp.10. Authors also should include a section where they clearly indicate which aspects of their algorithmic pipeline are original contributions and which are “off-the-shelf” solutions.

There is a confusion in lines 193-203. To my understanding (and according to fig.3) the color correction stage takes place at the end of the pipeline, but in line 193 it is stated that it takes place in the beginning. Which is correct? Please clarify that in the manuscript. In fig. 3 please retain the camera selection box, without information of the different techniques. It is a bad practice to mix alternatives of various stages in a high-level description of a processing pipeline.  Keep it to a more abstract level.

Why did not authors use a %error threshold in the depth consistency rule (eq. 14) and opted for an absolute error? To my experience the absolute error of measurement for RGB-D sensors is larger the further it is an object from the sensor? Isn’t it a more natural way to validate the depth in a way that respects the limitations of the RGBD sensors?

There should be more detailed formulation regarding the propagation algorithm (line335-6).

The experimental evaluation of the proposed scheme is rather limited and subjective. Is there any way to quantify the improvements of the proposed method to e.g. standard approaches that utilize only RGB information for the same dataset?

Author Response

“Authors propose a method for mapping texture into a 3D mesh, by utilizing RGB-D data in order to compensate for errors in camera-pose estimations that characterize regular RGB images that lead to unwanted visual artifacts. The proposed method consists of typical stages where the 3D mesh is projected onto the camera’s plane for each acquisition, and a z buffering stage is used to define which components of the mesh are visible. Consequently, a stage where the depth information is used to validate each mesh component, by comparing the estimated and measured depth, and a selection of the best camera for each triangle is performed via a propagation algorithm that promotes the selection of same cameras for neighboring triangles. The border faces (triangles that are neighboring to others with different cameras selection) are blended in order to smooth the transitions, and the images are color-corrected in a global optimization process that minimizes the inconsistency between the color of same faces as appear to different cameras. The paper is fairly structured and the language is good.”

We thank the reviewer for their analysis and positive feedback.

“The algorithmic flow seems straightforward, but I have doubts regarding the ease of reproducibility if authors do not provide source code in a public repository, since the overall algorithm contains numerous technicalities.”

As per the reviewer’s request, we have made the code openly available on Github: https://github.com/lardemua/rgbd_tm

We have added reference to this repository within the manuscript (line 246).

“Authors should make clear what they propose and what is going to be their contribution in the introduction of the paper, because in it’s current form is rather vague. The contribution becomes clear only on line 260, pp.10. Authors also should include a section where they clearly indicate which aspects of their algorithmic pipeline are original contributions and which are “off-the-shelf” solutions.”

This is a very good point. We have made an effort to better and more clearly identify the contributions in the introduction, as suggested by the reviewer. We believe this will contribute rather positively to the clarity of the writing. A new paragraph was added to the introduction (line 112):

“The core contribution of this paper is to propose a complete processing pipeline which, taking as input a mesh and a set of inaccurately registered images, is able to produce more visually appealing textures when compared with classic winner-take-all or average-based approaches. Additionally, there are various innovative features within the pipeline, from which we highlight: i) a propagation algorithm that ensures minimization of the transitions in the process of camera selection for each triangle in the mesh, which are responsible for creating visual artifacts; ii) the use of RGB-D images, as opposed to simply RGB images, for the purpose of texture mapping. The additional information provided by the depth channel is used to compute what we call the depth consistency cost; iii) a localized smoothing of texture in the regions of the mesh where there is a change in camera selection.”

“There is a confusion in lines 193-203. To my understanding (and according to fig.3) the color correction stage takes place at the end of the pipeline, but in line 193 it is stated that it takes place in the beginning. Which is correct? Please clarify that in the manuscript. In fig. 3 please retain the camera selection box, without information of the different techniques. It is a bad practice to mix alternatives of various stages in a high-level description of a processing pipeline.  Keep it to a more abstract level.”

We thank the reviewer for paying such close attention. To clear things up, we have added the following paragraph to the section “Global Color Adjustment” (line 400):

“Global color adjustment is an optional part of the pipeline. When used, this global color correction procedure adjusts the colors of all the images before the image selection, to texture map the triangles in the mesh. As such, it occurs as a pre-processing operation, before the texture mapping process.“

Additionally, we have altered Fig. 3 according to the reviewer’s specifications. We agree that this high-level description of the pipeline should be kept more abstract.

”Why did not authors use a %error threshold in the depth consistency rule (eq. 14) and opted for an absolute error? To my experience the absolute error of measurement for RGB-D sensors is larger the further it is an object from the sensor? Isn’t it a more natural way to validate the depth in a way that respects the limitations of the RGBD sensors?”

The reviewer is correct in asserting that the error of RGBD sensors is dependent on the range at which the measured object is. The reviewer’s suggestion is valid, still we believe that using absolute values is more helpful when parameterizing the algorithm. In any case, the mathematical formulation can be easily adjusted to both approaches.

“There should be more detailed formulation regarding the propagation algorithm (line335-6).”

As per the reviewer’s request, we have added a paragraph at the end of section “Camera Selection”, as follows (line 348):

“The propagation starts from a random face in the mesh and is carried out by visiting the neighboring faces of the previously visited faces. The algorithm annotates which faces have already been visited in order to cycle through the entire mesh, without revisiting any. Each time a face is visited, the camera selected to texture map it is the same as that of the neighbor triangle from which the visit was triggered, i.e. the parent triangle. Note that sometimes this is not possible because the visited triangle is not observed by the camera selected for the parent triangle. When this occurs, transitions in camera selection will be noted in the mesh.”

“The experimental evaluation of the proposed scheme is rather limited and subjective. Is there any way to quantify the improvements of the proposed method to e.g. standard approaches that utilize only RGB information for the same dataset?”

The reviewer is right in this regard. Most evaluations within this area of research have this limitation, and this is something we have also taken note of. This is why we are currently developing definitions for qualitative and quantitative mesh quality metrics. It is, however, no trivial task. In the future we would like to use these metrics to compare various techniques of mesh processing with the technique presented in this manuscript.

Reviewer 3 Report

It is a well-written and well-designed paper. 

The only problem is the lack of comparisons with numerical results to validate the approach.

For instance, from the text: "The proposed approach produces the best quality textures of all four approaches". However, the authors have shown just one part of the image that could be carefully selected to demonstrate the algorithm's superiority and ignoring more sensitive parts.

Moreover, the tests were also considering just one scenario with is not very challenging in terms of diversity.

I suggest using at least a toy sample (a virtual 3D scenario) to simulate the RGBD capture by using ROS and creating a quality index that can evaluate the entire scene.

Author Response

“It is a well-written and well-designed paper.” 

We thank the reviewer for their positive feedback.

“The only problem is the lack of comparisons with numerical results to validate the approach. For instance, from the text: "The proposed approach produces the best quality textures of all four approaches". However, the authors have shown just one part of the image that could be carefully selected to demonstrate the algorithm's superiority and ignoring more sensitive parts. Moreover, the tests were also considering just one scenario with is not very challenging in terms of diversity. I suggest using at least a toy sample (a virtual 3D scenario) to simulate the RGBD capture by using ROS and creating a quality index that can evaluate the entire scene.”

Our main goal in this work was to present our novel approach and validate the proposed processing pipeline. The acquisition and processing of scenes has been negatively impacted by the current pandemic. It is a pertinent suggestion by the reviewer, and something we have planned for the future, but we also believe that the proof of concept presented here is sufficient to demonstrate the relevance, contributions and main features of our work. From it, we have selected the information we deemed necessary to showcase the results, while trying not to clutter the manuscript with too many images of the scene. We would like to make explicit that we did not look for regions in which our approach worked well and the others did not. Moreover, we can state that, as far as we have observed, all the regions of the mesh hold the same average trend of results. 

There is another aspect of the reviewer’s comment which we also consider very relevant. That it that the comparison with other approaches would benefit from more quantitative metrics, as is the case with most evaluations in published works within this area of research, to our knowledge. This is something that has caught our attention, and as such, we are currently defining and implementing qualitative and quantitative mesh quality metrics. It is, however, no trivial task, and at the present moment there are no off-the-shelf solutions for such comparisons. In the future, we would like to use these metrics to compare various techniques of mesh processing with, and showcase, the technique presented in this manuscript.

Round 2

Reviewer 3 Report

the authors have provided a fair justification.